# Experimental Study on Deformation and Load Transfer Mechanisms of Symmetrical Batter Piles under Vertical Loading

**Kaiyuan Liu [1], Chao Han [2,\*], Chengshun Xu [2] and Zhibao Nie [3]**

[1]   Key Laboratory of Urban Security and Disaster Engineering of Ministry of Education,
     College of Architecture and Civil Engineering, Beijing University of Technology, Beijing 100124, China;
     liukygeo@163.com
[2]   College of Architecture and Civil Engineering, Beijing University of Technology, Beijing 100124, China;
     xcs_2017@163.com
[3]   Geotechnical Engineering Laboratory, China Electric Power Research Institute, Beijing 102401, China;
     niezhibao@epri.sgcc.com.cn
\*   Correspondence: Charleshan@emails.bjut.edu.cn

**Abstract:** Under the action of vertical loading, batter piles rarely appear individually, as they undergo horizontal and vertical displacements at the same time and produce a sizeable additional bending moment. However, previous studies have mainly focused on a single batter pile, which is inconsistent with engineering practices. Although single pile tests can easily reveal its working behavior, they also ignore two important factors, namely, the internal force redistribution caused by the deformation limitation of the mirror-like pile, and the interaction between the symmetrical piles and "clamped" soil (the soil between two symmetrical piles). Therefore, this paper took symmetrical batter piles as the test object to explore the influence of the two factors on the load transfer mechanism. Moreover, the deformation mode, distribution of inertial forces, and group effect of symmetrical batter piles were also discussed. The results showed that the "clamping effect" caused by the pile deformation had a significant impact on the load transfer. Under vertical loading, the flexible symmetrical floating batter piles were the only deformation mode. Under the constraint of the cap and mirror-like batter piles, the symmetrical conformation partially compensated for the disadvantage of the additional bending moment.

**Keywords:** symmetrical batter piles; vertical loading; tests; deformation mode; load transfer mechanism

## 1. Introduction

Batter piles are widely used in infrastructure construction. Its high impedance to the lateral load makes it irreplaceable in the design of bridges, wharves, and transmission towers [1]. They also have disadvantages, as it is generally accepted in academia that under vertical loads, batter piles will produce a sizeable additional bending moment and be accompanied by the horizontal and vertical displacements [2,3]. Therefore, some codes, such as Eurocode-8 [4], do not recommend using batter piles in seismic-prone areas. Other scholars believe that the insufficient understanding of the batter piles' working behavior and improper design are the main reasons for the occurrence of damage [5]. If properly designed, the batter pile can be beneficial to both the superstructure and itself [6–8]. The argument about whether the usage of batter piles is detrimental or beneficial is still unsettled. Its working behavior and load transfer mechanism are more complex than those of plumb piles.

Meyerhof [9,10] was the earliest to conduct a series of tests on single batter piles. The bearing capacity of the piles was found to depend on the layered structure, load inclination, and pile batter. Modified equations for displacement [11] and the empirical formula for the ultimate bearing capacity under arbitrary inclined loads [12] were proposed. As the pile

inclination range was small, the method was obtained from the modification of the bearing capacity of the plumb pile. However, it is not suitable for piles with large inclinations, which are increasingly being used widely in engineering [13]. Rao et al. [14] considered the pile–soil separation and proposed an empirical formula that can predict the additional bending moment of a single batter pile caused by soil settlement. The comparison between the analytical and test results showed that the settlement would introduce a sizeable additional bending moment, which may lead to the failure of the batter pile.

In a centrifuge test, Zhang et al. [15,16] found that the vertical load had different influences on the lateral resistance of positive and negative batter piles. When a vertical load was applied, the lateral resistance of a negative battered pile was reduced because of the bending moment due to the vertical load component that causes an additional lateral displacement. The opposite was true for a positive battered pile as the bending moment counterbalance the lateral load component. However, the constraint of the cap and mirror-like batter was not considered in the test. This constraint may result in the redistribution of the internal force. In fact, it was found that in this paper, the symmetrical conformation could partially compensate for the disadvantage of the additional bending moment, thus reducing the concentration and maximum value of the additional bending moment. Moreover, batter piles rarely bear vertical load individually, which is inconsistent with engineering practice. Cao et al. [17] investigated the influence of pile inclination on the single batter piles' working behavior by laboratory tests. The results showed that the greater the inclination, the more rapidly the axial force decays. The maximum shear force appeared at the pile head and increased with the pile inclination. However, the test did not consider the mirror-like batter pile either, which is different from the engineering application.

Mroueh et al. [18] explored the bearing capacity of a batter pile under the combination of vertical and horizontal loading by the finite element method. The relationships between the bearing capacity, the loading, and the pile inclination were given in a chart. However, the axial load caused the stiffness degradation of soil around the pile [19], which could not be reflected by the Mohr–Coulomb constitutive model. The simple Coulomb friction criterion was used at the pile–soil interface; thus, the model is too simple to reflect the truth. Mohab et al. [20] obtained the critical pile inclination of batter piles by analyzing the test and numerical results. They believe that when the pile inclination is less than $30°$, the axial bearing capacity will change slightly, whereas when the inclination is greater than $30°$, the axial bearing capacity will decrease with the pile inclination. Hanna et al. [21] studied the influence of pile inclination on the skin-friction of batter piles through laboratory tests. However, they regarded the skin-friction in the entire pile as the same. The variation in skin-friction along the depth was not considered, which is inconsistent with the truth.

In summary, previous studies have mainly focused on the bearing capacity of batter piles. The working behavior, deformation mode, internal force distribution, and development under vertical loading have rarely been reported. The relevant tests are necessary. When subjected to vertical loading, horizontal and vertical displacements of the batter pile will occur simultaneously. Two important factors have been ignored in previous studies: First, the constraint of the pile cap and mirror-like batter; second, the interaction between the symmetrical batter and the soil clamped by the pile, namely, the "clamping" effect. All of the above two factors will introduce a redistribution of internal force and make the working behavior more complex. Therefore, symmetrical batter piles were set as the test object in this paper. The main purpose was to further understand the load transfer mechanism and the difference from the single pile.

## 2. Design and Implementation of the Test

### 2.1. Design of the Test

In this paper, five groups of tests were carried out. The variables were pile spacing $D$ and pile inclination $\beta$ (angle between pile axis and mud-line), as shown in Table 1. The definition of parameters is shown in Figure 1, where $e$ is the loading height, $L$ is the length

of the pile, and $E_P$ and $G_S$ are the elastic and shear moduli of the pile and soil, respectively. The direction of pile inclination is called the front of the pile, and the opposite is called the back of the pile.

**Table 1.** Grouping of the tests.

| Grouping Number | Pile Inclination $\beta°$ | Pile Spacing | Loading | Notes |
|---|---|---|---|---|
| TDP1 | 10° | — | | $D$ = 40 mm |
| TDP2 | 10° | 3D | Multistage | Loading height $e$ = 2.5 $D$ |
| TDP3 | 10° | 5D | loading- | $L/D$ = 27.5 |
| TDP4 | 20° | 5D | vertical↓ | Embedded Depth = 1 m |
| TDP5 | 10° | 7D | | |

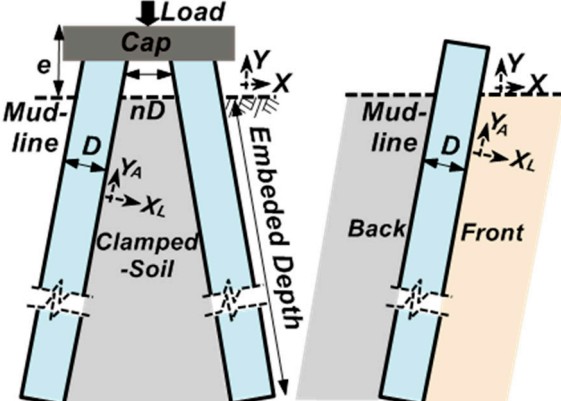

**Figure 1.** The definition of parameters and local coordinate system.

Poulos [22] classified piles as rigid and flexible according to the coefficient $Kr$, as shown in Equation (1), where $E_pI_p$ is the bending stiffness of the pile section, $L_p$ is the embedment depth, and $E_s$ is the tangent modulus of soil. In this paper, $E_pI_p$ = 1.98 kNm² and $Kr$ = 2 × $10^{-4}$. The batter pile will be discussed in the frame of flexible piles. Figure 2 shows the conformation and dimension of the model pile.

$$K_r = \frac{E_p I_p}{E_s L_p^4} \tag{1}$$

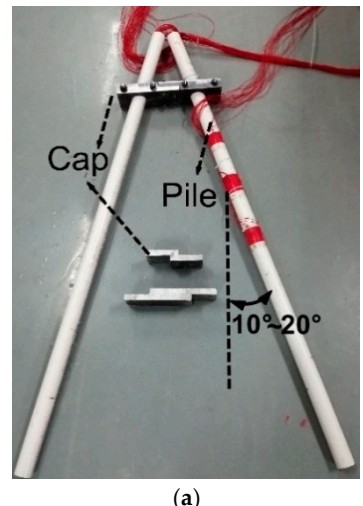

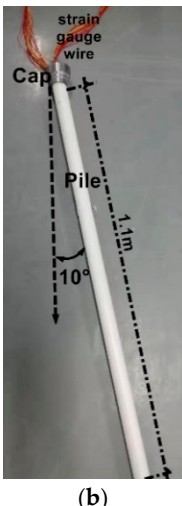

(a)

(b)

**Figure 2.** The diagram of the model pile. (**a**) The symmetrical batters and (**b**) the single batter pile.

## 2.2. Materials

The batter piles were made of thick-walled PVC pipes with an outer diameter of 40 mm and a length of 1100 mm. The elastic modulus measured by the three-point bending test was 3.3 GPa. According to Wood [23], the similarity law and parameters are shown in Table 2.

**Table 2.** Similarity design.

| Parameters | Model | | Scale Factor | | | Unit |
|---|---|---|---|---|---|---|
| | Law | Parameter | Prototype | Actual Value | Actual Coefficient | |
| Depth | $\lambda$ | 0.08 | 13.75 | 1.1 | 0.08 | m |
| Density | $\eta$ | 1 | 16.4 | 17.2 | 1.05 | t/m$^3$ |
| Pile L | $\lambda$ | 0.08 | 13.75 | 1.10 | 0.08 | m |
| Diameter | $\lambda$ | 0.08 | 0.5 | 0.04 | 0.08 | m |
| E | $\frac{I_M(1/\lambda)^{4.5}}{I_P}$ | 0.29 | $25 \times 10^9$ | $3.3 \times 10^9$ | 0.132 | Pa |

(EI)P and (EI)M are the bending stiffnesses of the prototype and model, respectively, $\lambda$ is the geometric similarity ratio.

Dry river sand was used in the test with $Cu$ = 6.43 (coefficient of uniformity), $Cc$ = 1.27 (coefficient of curvature), and $D_{50}$ = 0.503 mm (average particle size). The internal friction angle was measured by a direct shear test under the same density of $\phi = 38°$.

## 2.3. The Testing Equipment and Sensors

The equipment and loading method of the test are shown in Figure 3, where Figure 3a shows the test photo, Figure 3b shows the arrangement of sensors, and Figure 3c shows the cap with a variable pile spacing.

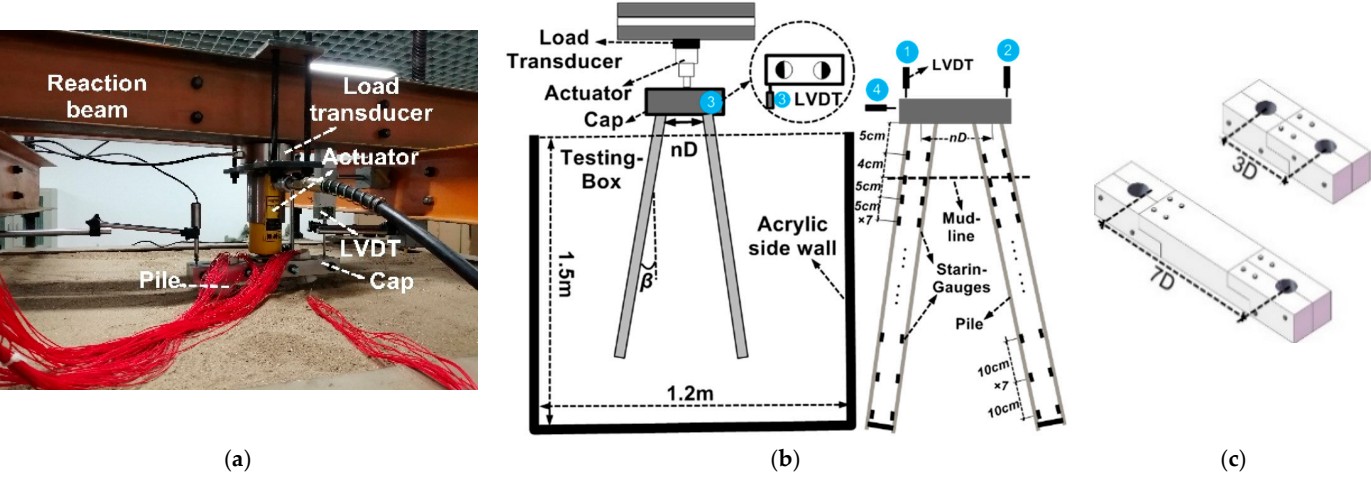

(a)          (b)          (c)

**Figure 3.** Equipment and sensors of the test: (**a**) testing photo, (**b**) the arrangement of sensors, and (**c**) the cap with a variable pile spacing.

The test was carried out in a self-balancing testing box with dimensions of 1.5 m × 1.2 m × 1.5 m (length × width × height). The minimum distance between the pile and the acrylic sidewall was greater than 15*D*, so the boundary effect could be ignored [24]. When applying the load, the tail of the actuator was against the force transducer, and the force transducer was fixed on the reaction beam, which was rigidly connected to the testing box to form a self-balancing loading device.

The layout of strain gauges is shown in Figure 3b. The pile was divided into two parts from the middle. Strain gauges were attached to each part along the axial direction. There were 15 pairs of strain gauges on the whole pile, and the first pair was 5 cm above the

mud-line. The strain gauges 0–30 cm below the mud line were densified with a spacing of 5 cm, and the remainder were densified with a spacing of 10 cm. When the epoxy resin was cured, the pile was reclosed with special adhesive. Cable ties were settled every 100 mm along the pile axis to increase the overall strength. The pile end was closed with a pipe plug. The pile surface was brushed with glue and rolled in the sand to simulate the working condition of the bored pile [25].

Four LVDTs (Linear Variable Differential Transformer) were settled at the side and top of the cap to measure the vertical (No.1 and No.2) and lateral displacements (No.3 and No.4), as shown in Figure 3b. When the output difference between the two vertical LVDTs was greater than 5% or the lateral displacement was greater than 5 mm, the test was repeated.

The cap was composed of 4-piece steel parts (Figure 3c). The holes were brushed with special adhesive and fastened with bolts to simulate the fixed pile head. Prefabricated parts could be connected between the clamps to simulate different pile spacing.

### *2.4. Test Procedures*

The bearing stratum was filled to 50 cm five times, controlling the weight and filling height each time. Each layer was compacted with a vibrator for the same length of time under 50 Hz–360 kW. The average density of sand was 1.72 g/cm$^3$ and the relative density was 66%. The symmetrical batter pile was placed by a fixture before it could self-stabilize.

A multistage loading was carried out. According to the Chinese Code JGJ106-2003 [26] and Ayothiraman et al. [27], when the load is below 300 N, the increment is 50 N. When the load is within 300–1000 N, the increment is 100 N. If the value of each LVDT is less than twice the value of the previous stage and converges within 5 min, the next stage is applied after 10 min. Otherwise, the loading is stopped. Moreover, the displacement limit is set to a maximum of 1.2 cm (0.3*D*) [28].

## 3. The Deformation Mode and Bearing Capacity of Symmetrical Batter Piles

The deformation along the pile depth is an important aspect of the working behavior. This section first analyzes the deformation mode of symmetrical batter piles under vertical loading. On this basis, the load–displacement curve is corrected. The strain scatters of $(-\Delta\varepsilon(z))/d$ are fitted directly, and then quadratic integration is performed to obtain the pile deformation displacement in the $X_L$-$Y_A$ coordinate system, as shown by Equation (2). In this way, the nonlinearity of the pile material can be considered, and errors introduced by the bending moment fitting and differentiation can be avoided, where $\Delta\varepsilon$ is the difference between a pair of strain gauges and *d* is the inner diameter of the pile.

$$y(Z) = \int_0^Z (\int_0^Z -\frac{\Delta\varepsilon(Z)}{d}dz)dz + C_1 Z + C_2 \qquad (2)$$

Figure 4 shows the mean value of the deformation along the depth. All test results in the $X_L$-$Y_A$ coordinate system are shown in Figure 4a,b, which is the pile deformation with the same spacing and different inclination $\beta$. Figure 4c shows the pile deformation with the same inclination $\beta$ and different spacing *D*. Figure 4d shows the true deformation model of the symmetrical batter pile in the X-Y coordinate system. The result in the next section shows that when failure occurs, the maximum pile tip resistance is within 20% of the applied loading. Therefore, the deformation mode described here can be considered as the mode of floating symmetrical batter piles with a fixed pile head.

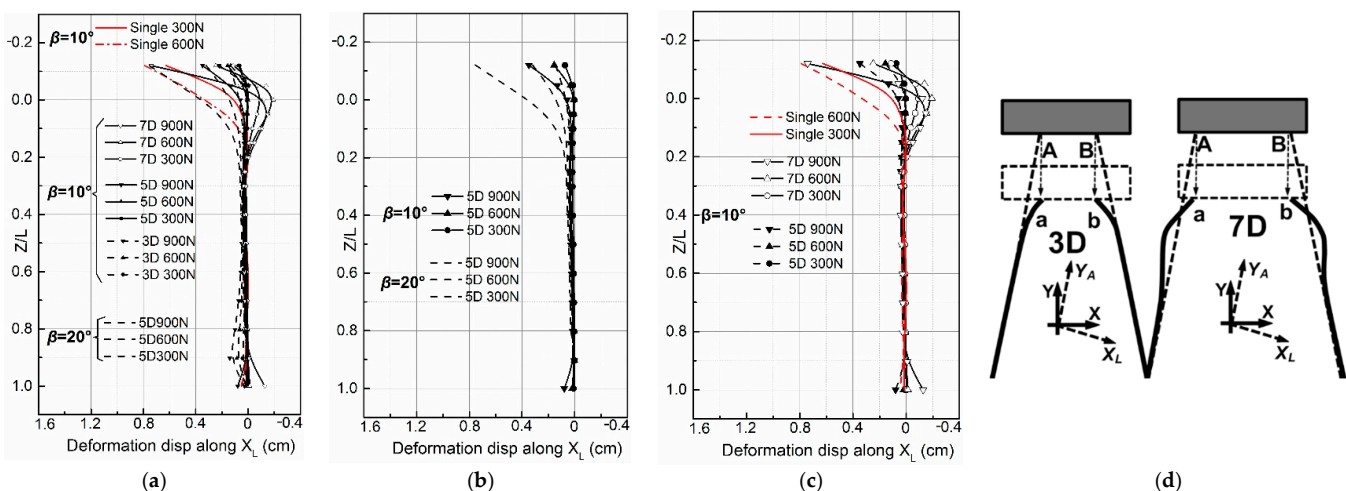

**Figure 4.** Deformation displacement of the batter pile along depth. (**a**) The deformation of each test, (**b**) 3D $\beta = 10°$ vs. $\beta = 20°$, (**c**) $\beta = 10°$ 3D vs. 7D, and (**d**) the true deformation mode in X-Y coordinate system.

It can be seen from Figure 4a that the deformation mainly occurs in the upper pile section ($\leq 0.2L$). As shown in Figure 5b,c, the maximum deformation value increases with the pile inclination $\beta$ and pile spacing $D$. When the pile inclination $\beta$ changes from $10°$ to $20°$ or the pile spacing increases from $5D$ to $7D$, the deformation value increases by more than twice. When the pile spacing increases to a certain value ($7D$ in this paper), negative deformation displacement appears, as shown in Figure 4c.

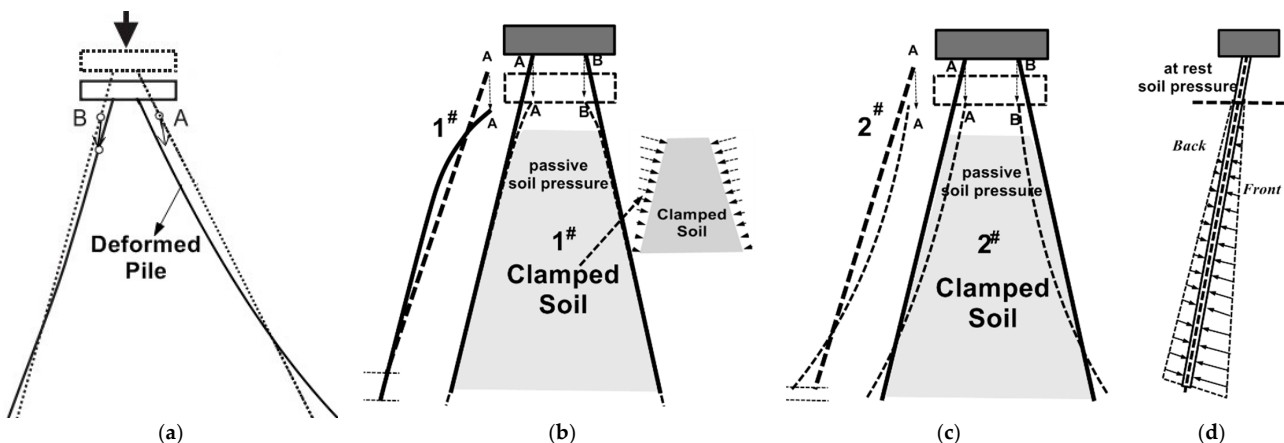

**Figure 5.** The assumption of the deformation mode of the symmetrical batter pile under vertical loading. (**a**) Zhang et al.'s study, (**b**) deformation assumption No.1, (**c**) deformation assumption No.2, (**d**) the at-rest soil pressure in the front and back areas of the batter pile.

There are two coordinate systems in Figure 5d. The X and Y are parallel to the mud-line and gravity direction. $X_L$-$Y_A$ is parallel to the lateral and axial directions of the pile. The deformation derived from strain occurs in the $X_L$-$Y_A$ coordinate system (Figure 5a–c). It should be noted that in the X-Y coordinate system, the boundary conditions at the joint between the pile and cap only allow the pile head to have displacement in the Y direction. Therefore, considering the boundary conditions and angular relationship, the true deformation mode in X-Y is shown in Figure 4d.

In the tests, only one deformation mode is obtained (although the 7d test has negative deformation, it is essentially the same as the small pile spacing test group); namely, when the pile spacing is small, the upper batter pile section bends inward to compensate the displacement in the X-direction. With the increase in the pile spacing, the deformation develops deeper and is accompanied by the outward bulging deflection. The author

believes that when subjected to the vertical loading, the flexible symmetrical floating batter piles has only one kind of deformation mode. The reasons are illustrated in Figure 5.

Figure 5a shows the deformation mode assumed by Zhang et al. [29]. The author believes it has limitations. When the symmetrical batter pile is subjected to the vertical loading, the settlement difference between the cap and pile tip must be compensated by the pile deformation. Due to the fixed pile head, the displacement along the X direction is not allowed. Therefore, only two kinds of deformation modes can meet the boundary conditions of the pile and cap connection. They are shown as Figure 5b,c, called mode No. 1 and No. 2, respectively. The deformation of No. 2 is the same as the assumption of Zhang et al.

Due to the pile inclination and dead weight of the pile, the at-rest soil pressures in the front and back areas of the batter pile are different (Figure 5d) [21]. The pile front is higher than the pile back. From the perspective of force balance, deformation mode No. 2 is less likely to occur. Second, the "clamping" effect makes No.2 more difficult to occur. The "clamping" refers to the compression of soil between the piles caused by the deformation. This phenomenon is consistent with Li's report [30]. The symmetrical batter pile deforms in opposite directions. The greater the pile deforms, the greater the confining pressure on the trapezoidal soil (Figure 5b) between the two piles. It is widely known that the increase in confining pressure will decrease the void ratio and increase the soil stiffness. Considering the above two reasons, only deformation No. 1 could occur, and it is also the only kind of deformation mode. Moreover, the "clamping" effect will enhance the pile–soil interaction. The variation in axial force, pile tip resistance ratio, and skin-friction are also affected. These issues will be discussed in the next section. For very flexible piles such as micro-piles, and batter piles of the high-piled wharf and super-long batter piles, the deformation mode mentioned above should be given sufficient attention.

According to the deformation value of the batter pile, the method in Figure 6 was used to correct the displacement of the cap based on Equations (4) and (5). Figure 6a,b show the correction method corresponding to deformations No.1 and No.2, respectively. Figure 7 is the corrected result. The response of the pile cap only considers the factors of skin-friction and pile tip resistance, and the influence of deformation ($\Delta L$) is removed. The ultimate bearing capacity is also listed in the lower left corner.

$$\Delta L = L(1 - \cos\theta) \tag{3}$$

$$\Delta L = \Delta L_1 + \Delta L_2 = L_1(1 - \cos\theta_1) + L_2(1 - \sin\theta_2) \tag{4}$$

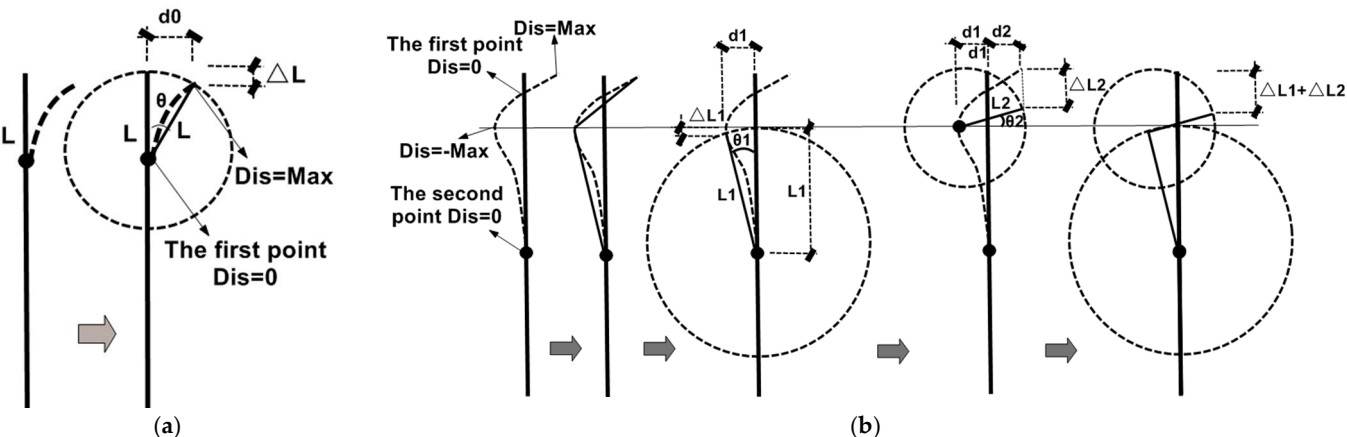

**Figure 6.** Displacement correction method of batter piles (**a**) No.1 and (**b**) No.2.

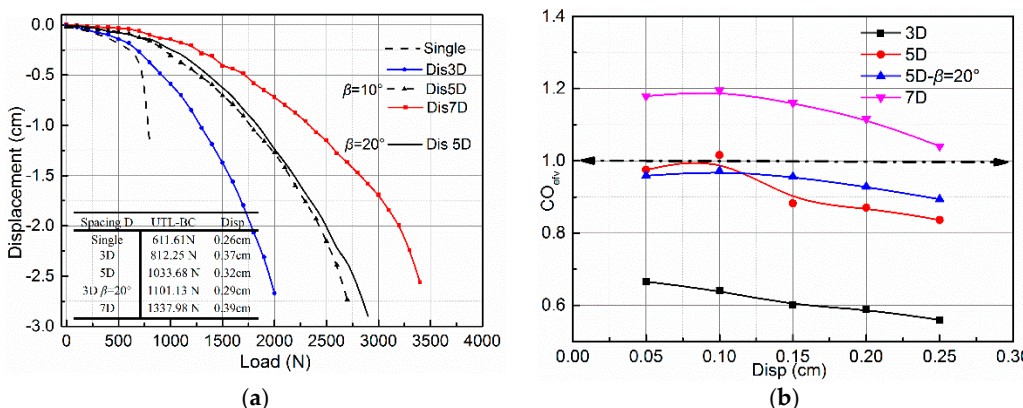

**Figure 7.** (**a**) Load–displacement of pile cap, and (**b**) variation in symmetric batter pile group efficiency with pile spacing.

It can be seen from Figure 7a that when the pile inclination $\beta$ changes from 10° to 20°, the differentiation between the load–displacement curves is slight. The relative difference is less than 5%, which is quite different from the single pile test (the bearing capacity increases with the pile inclination). The author believes that this is due to the x-direction constraint weakening the influence of the pile inclination on the ultimate bearing capacity.

Although the pile spacing increases with the depth, the pile group effect is still obvious from the difference in ultimate bearing capacity between different pile spacings. In this paper, the test with a pile spacing of 7$D$ was set as the reference to investigate the influence of pile spacing and inclination on the group effect.

The pile group effect is defined in Equation (5), where $UV_{nD}$ and $UV_{\text{single}}$ represent the loads of the symmetrical and single batter pile under the same displacement, respectively (below the failure load), and $n$ is the number of piles in the pile group, which equals 2 in this paper. The lower the coefficient, the more significant the pile group effect. The results are shown in Figure 8b.

$$CO_{efV} = \frac{UV_{nD}}{n \times UV_{\text{single}}} \tag{5}$$

It can be seen from Figure 8b that the increase in inclination and pile spacing will all reduce the pile group effect, but the influence of pile spacing is more significant. When the pile spacing is 3$D$, the coefficient is only half of that when the spacing is 7$D$. The increase in pile inclination $\beta$ helps to reduce the stress superposition area in soil. Therefore, when the inclination changes from 10° to 20° under the same pile spacing, the coefficient $CO_{efv}$ decreases by about 7%. In most codes, the critical pile spacing of plumb pile without the group effect is between S = 6$D$ and 8$D$ (Sandy soil). For example, the critical longitudinal pile spacing in American API code [31] is 8$D$. The pile foundation code of Poland [32] is 6$D$. For batter piles, the pile inclination will reduce the stress superposition area along the depth, thus reducing the pile group effect. In this paper, the pile group effect coefficient was already close to 1 when the pile spacing was 5$D$. When the spacing was 7$D$, the coefficient was greater than 1. Therefore, the pile group effect of the batter pile foundation should be modified considering the influence of inclination.

## 4. The Internal Force Distribution of Symmetrical Batter Piles and "Clamping" Effect

The distribution and variation in axial force, pile tip resistance ratio, skin-friction, and the influence of the "clamping" effect on them will be analyzed in this section.

Figure 8 shows the variation in axial force along the depth of the batter pile, where Figure 8a shows the axial force of all tests. Figure 8b shows the variation in axial force under the same pile inclination and different pile spacings, while Figure 8c shows the variation in axial force under the same pile spacing and different pile inclinations. It can be seen that the axial force decreases nonlinearly along the depth and presents as an inverted

triangle, which is similar to the distribution of the plumb pile. It indicates that the activation sequence of the skin-friction is identical to the plumb pile.

However, it can be seen from Figure 8a,b that in the upper pile section (the dotted circle), the attenuation rate of axial force gradually increases with the pile spacing. This tendency is more obvious when the load is greater, which is quite different from the symmetrical plumb pile. The author believes that this is caused by the "clamping" effect. The deformation of symmetrical batter piles leads to the compression of the soil between the piles. The pile–soil normal contact force increases and so does the friction coefficient. Moreover, the increase in confining pressure will result in the increase in the soil stiffness. All the reasons together will lead to an increase in the skin-friction attenuation rate. In terms of Figure 4, the greater the pile spacing, the larger the deformation of the upper pile part, which will further aggravate the "clamping" effect. This is consistent with the phenomenon that the skin-friction attenuates rapidly in the upper part of the pile with greater pile spacing.

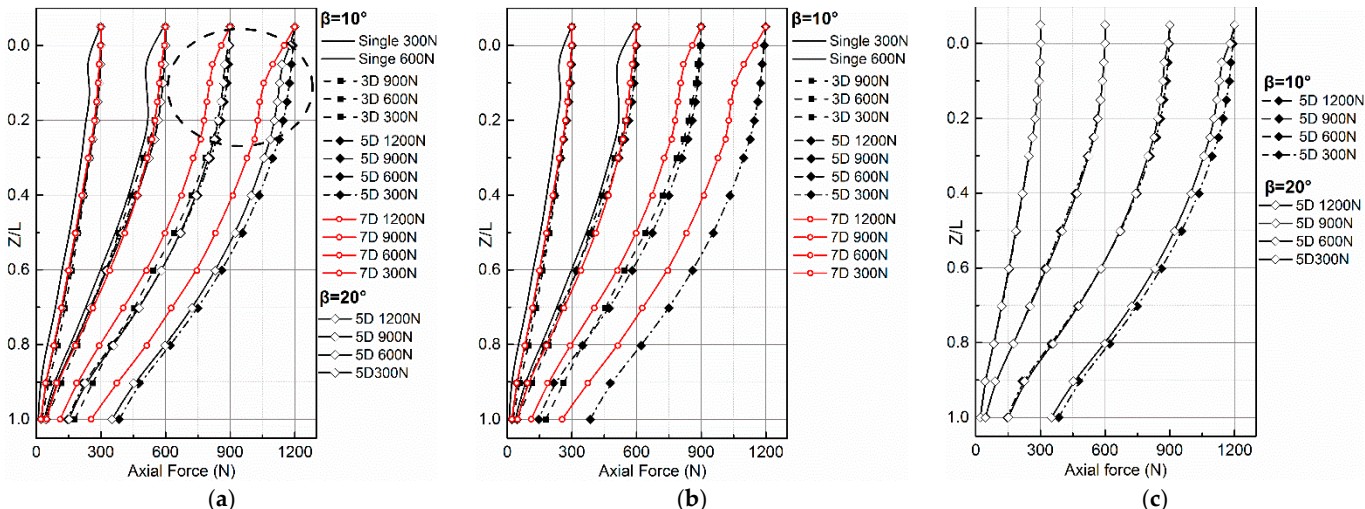

**Figure 8.** The variation in axial force along depth: (**a**) axial force of all tests, (**b**) axial force when $\beta = 10°$, and (**c**) axial force $\beta = 10°$ vs. $\beta = 20$.

Figure 9c shows the influence of pile inclination on axial force. The influence area also appears on the upper part. The working principle is the same as the influence of the pile spacing on axial force, namely, the deformation of the upper section of the batter pile enhances the "clamping" effect. However, the influence of the pile inclination on the axial force is less than that of the pile spacing, and it is significant only when the load is greater.

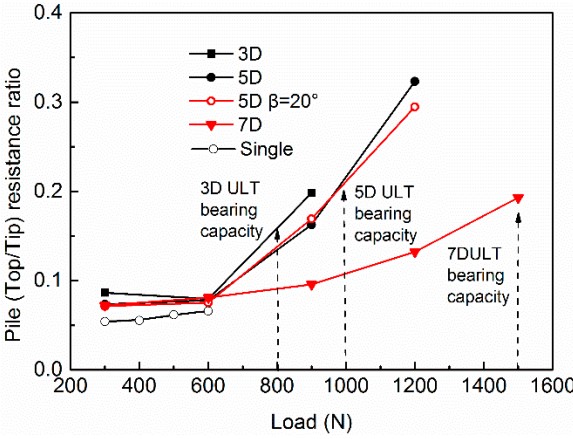

**Figure 9.** The variation in the pile tip resistance ratio of the symmetrical batter pile.

The pile tip resistance ratio (the ratio between the pile head axial force and pile tip reaction force) is an important indicator for investigating the load transfer of the pile foundation [33]. It is commonly used on plumb piles subjected to vertical loading, which can directly reflect the proportion of loading borne by the pile tip and skin-friction.

Figure 9 shows the variation in the pile tip resistance ratio under different loadings. The position indicated by the dashed arrow is near the ultimate bearing capacity of each test. It can be seen that when the loading is within 300–600 N, the pile tip resistance ratios are the same. This indicates that the pile skin-friction along the pile axis is not fully activated. As the load increases, the pile tip resistance shows a nonlinear increase. The ratio is negatively correlated with the pile spacing.

In terms of Figures 5 and 6, the stronger the "clamping" effect, the earlier and more sufficient the skin-friction is mobilized. Therefore, the load transferred to the pile tip is reduced, which indirectly affects the pile tip resistance ratio.

The averaged skin-friction per unit length of the batter pile can be obtained from Equation (6), where $Q_i$ is the axial force and $Z_i$ is the depth corresponding to $Q_i$.

$$f_{AS} = \frac{Q_i - Q_{i+1}}{Z_{i+1} - Z_i} \tag{6}$$

Figure 10 shows the variation in averaged skin-friction along the depth of each test, where the solid line represents the skin-friction near the failure load. It can be clearly seen from the figure that the skin-friction is divided into the deformation-affected zone and -nonaffected zones. The demarcation line depth is about 0.2–0.3*L*.

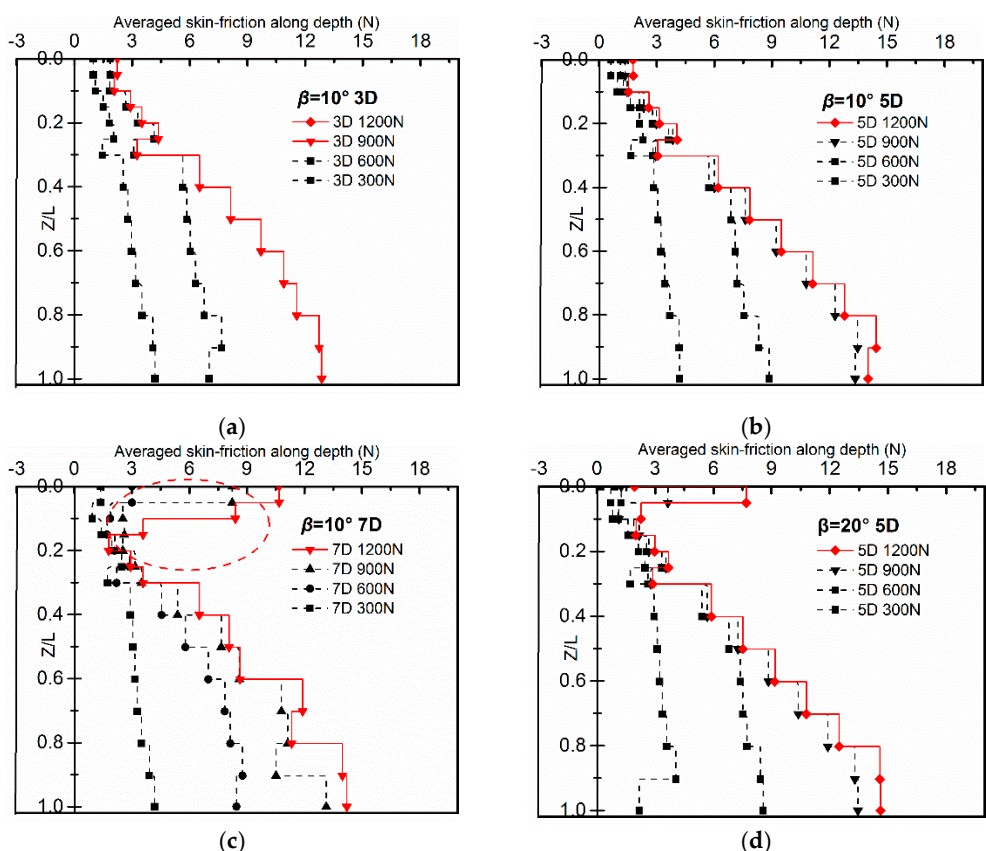

**Figure 10.** The variation in averaged skin-friction along depth: (**a**) *β* = 10° pile spacing = 3*D*, (**b**) *β* = 10° pile spacing = 5*D*, (**c**) *β* = 10° pile spacing = 7*D*, and (**d**) *β* = 20° pile spacing = 5*D*.

In the nondeformation-affected zone, the skin-friction remains constant when it reaches the limit value, which is similar to that of plumb piles. However, in the deformation-

affected zone, the limit value of skin-friction changes greatly due to the "clamping" effect. It increases significantly with the pile spacing and inclination, especially in Figure 10c,d. The phenomenon is consistent with the working principle of the "clamping" effect discussed above.

## 5. The Distribution and Development of the Bending Moment of Symmetric Batter Pile

Batter piles will generate additional bending moments under the action of dead weight or vertical load. This working behavior is significantly different from that of vertical piles, which is one of the main shortcomings pointed out by the community. In engineering practice, the conformation of symmetrical batter piles is widely used in wharves to bear the vertical loads of equipment and cargo. Therefore, it is necessary to explore the distribution of bending moments. The bending moment is obtained by Equation (7), where *EI* is the bending stiffness of the pile section, $\Delta \varepsilon$ is the difference between a pair of strain gauges, and *d* is the inner diameter of the pile.

$$M(T) = \frac{EI\Delta\varepsilon}{d} = \frac{EI(\varepsilon_1 - \varepsilon_2)}{d} \tag{7}$$

Figure 11 is the distribution of bending moments along the depth under vertical loading, where Figure 11a shows the bending moment of all tests and Figure 11b,c show the comparison of bending moments when the pile inclination or pile spacing is the same. It can be seen that regardless of how the pile spacing or pile inclination changes, the distributions are similar. The maximum bending moment appears at the pile head, and the attenuation rate below 0.2*L* is the same. When the pile spacing is 7*D*, the maximum bending moment of the pile head is largest for all tests. This is because the upper deformation of the pile is largest in all tests, thus enhancing the P-Δ effect.

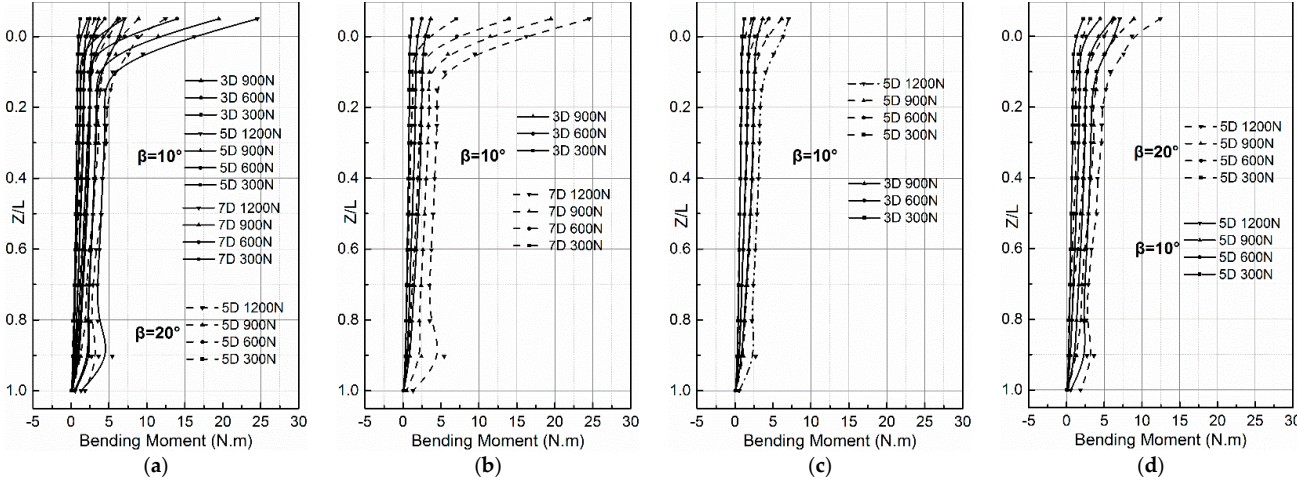

**Figure 11.** The variation in bending moment of the symmetrical batter pile under vertical loading: (**a**) the bending moment of all tests, (**b**) $\beta = 10°$ pile spacing = 3*D* vs. 5*D*, (**c**) $\beta = 10°$ pile spacing = 3*D* vs. 7d, and (**d**) pile spacing = 5*D* $\beta = 10°$ vs. 20.

The additional bending moment of batter piles is the major disadvantage when comparing with plumb piles. However, when the batter pile appears in the conformation of symmetry, this disadvantage is not that prominent. The comparison of the bending moment between the symmetrical and single batter pile will support the conclusion. The experimental results of Cao et al. [17] are also analyzed. (Cao et al. [17] carried out a series of tests of a single batter under vertical loading. There are two testing groups, in which the material and dimension of the pile, embedded depth, loading height, and method are the same as in this paper.)

Figure 12 shows the comparison results, where Figure 12a shows the test results of

this paper with different loadings (only loads below 600 N are shown, because 600 N is close to the failure load of the single batter pile), and Figure 12b shows the comparison between Cao and this paper under the load close to failure. It should be noted that in Cao's experiment, the free length is basically zero, so only the bending moment below the mud-line is analyzed.

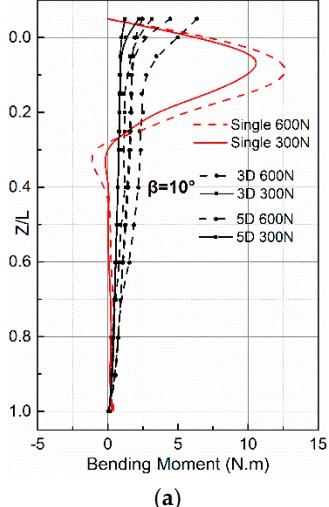
(**a**)

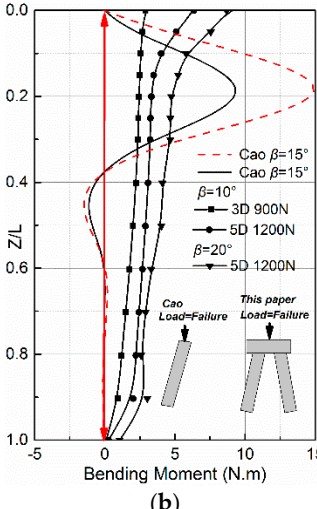
(**b**)

**Figure 12.** The comparison of bending moments between the single and the symmetrical batter piles under vertical loading. The comparison between the symmetrical and the single batter piles of (**a**) this paper and (**b**) Cao [17], under the failure load.

The disadvantage of an additional moment is fully displayed in the moment distribution of the single pile, namely, the large value and concentration of the bending moment. Due to the action of the free pile head and lateral load component, the P-Δ effect is enhanced, which results in the distribution of the bending moment under vertical loading being similar to that of the plumb pile under horizontal loading. Compared with the single batter pile, the symmetrical batter pile can activate deeper pile sections to resist loads either in the working state (Figure 12a) or under ultimate load (Figure 12a). The bending moment develops along the entire pile. Moreover, the P-Δ effect is reduced by the constraint of the pile cap and mirror-like pile. Therefore, the maximum bending moment decreases greatly. Even if the pile inclination of the symmetrical conformation is larger than that of the single pile, the maximum bending moment is reduced by 43% ($\beta = 20°$ compared with $\beta = 15°$). In conclusion, under the vertical loading, the symmetrical batter pile can partially compensate for the disadvantage of additional bending moments and weakens the concentration.

## 6. The Load Transfer Mechanism of Symmetric Batter Piles under Vertical Loading

The load transfer mechanism of flexible symmetrical floating batter piles under vertical loading is summarized in Figure 13. Due to the constraints of the cap and the mirror-like batter pile, the deformation of the symmetrical batter pile appears as an inward bending of the upper section. The deformation causes the soil between piles to be compressed and result in the "clamping" effect. The "clamping" effect will increase the attenuation rate of the axial force and the limit value of the skin-friction, which indirectly affects the pile tip resistance ratio. Moreover, the symmetrical conformation can weaken the additional bending moment effect and reduce the concentration and maximum of the bending moment. Therefore, in engineering practice, the difference between symmetrical and single piles and the influence of "symmetry" on the bearing capacity and load transfer should be fully considered.

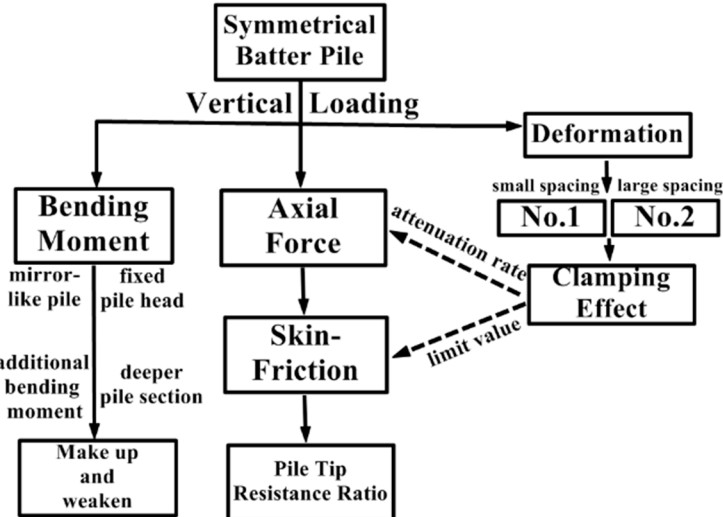

**Figure 13.** The load transfer mechanism of symmetrical batter piles.

## 7. Conclusions

The main conclusions of this paper are as follows.

(1) Under the vertical loading, there is only one kind of deformation mode of the flexible floating symmetric batter pile. When the pile spacing is small, the upper section of the pile bends inward. With the increase in the pile spacing, the deformation develops deeper and is accompanied by the outward bulging deflection.

(2) The greater the deformation, the more significant the soil "clamping" effect. The attenuation rate of the axial force and the limit value of skin-friction will increase, and the pile tip resistance ratio will decrease under the influence of the "clamping" effect.

(3) Under vertical loading, the additional bending moment caused by pile inclination is the main disadvantage of batter piles. Due to the constraint of the cap and mirror-like pile, the symmetrical conformation can reduce the influence of P-Δ and can partially compensate for the disadvantage of the additional bending moment. Moreover, the maximum value and concentration of the bending moment can also decrease, and the deeper pile section is activated to resist loads.

When the pile spacing is the same, the group effect of batter piles is smaller than that of plumb piles due to the pile inclination, and it becomes more significant with the increase in the pile inclination. At present, no study in the literature has given a method to describe the group effect of batter piles. However, based on experiments, using numerical and analytical methods, the pile group effect formula of plumb piles can be modified to describe the characteristic of the group effect of batter piles. This is also the focus of the author's future studies.

**Author Contributions:** K.L. conducted the experiments; C.H. wrote the draft of the paper; C.X. analyzed and processed the data; Z.N. helped improve the paper in organization and discussion based on his professional background in pile foundations. All authors have read and agreed to the published version of the manuscript.

**Funding:** The study is supported by the National Natural Science Foundation of China (No. 51178385 and No.51578026).

**Institutional Review Board Statement:** Not applicable.

**Informed Consent Statement:** Not applicable.

**Data Availability Statement:** All data included in this study are available upon request by contact with the corresponding author.

**Conflicts of Interest:** The authors declare no conflict of interest.

## Abbreviations

| | | | |
|---|---|---|---|
| $\beta$ | The pile inclination | The Subscript $p$ | The parameter of the prototype pile |
| $L/D$ | The pile length/pile diameter | The Subscript $M$ | The parameter of the model pile |
| $Kr$ | The flexible coefficient of the pile | $Cu$ | The coefficient of uniformity of sand |
| $EI$ | The bending stiffness of the pile section | $Cc$ | The coefficient of curvature of sand |
| $Es$ | The tangent modulus of soil | $D_{50}$ | The average particle size of sand |
| $I$ | The area moment of inertia | | |

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
