# Peer review of "Experimental Study on Deformation and Load Transfer Mechanisms of Symmetrical Batter Piles under Vertical Loading"

_applsci, doi:10.3390/app11073169_

Round 1
Reviewer 1 Report
The current study investigates several characteristics such as deformation mode and distribution of inertial forces and load transfer mechanism of symmetrical batter are evaluated experimentally. The authors conclude that clamping have significant influence on load transfer mechanisms and that symmetrical floating batter have single deformation mode. The overall aim here is to understand the load transfer mechanism and the difference from the single pile.
Please consider reviewing the abstract and highlight the novelty, major findings and conclusions.
What is the research gap did you find from the previous researchers in your field? Mention it properly. It will improve the strength of the article.
Line 15 instead of current it is better to say previous.
Why did the authors use the 10- and 20-degrees inclination in Table 1? Are they common standard angles or based on past literature or industrial applications or something else?
The table have many symbols and letters please create a list of symbols and nomenclature and add it at the start or end of the manuscript
Figure 2 is not necessary and can be removed, it is more suitable for a book chapter or literature review chapter in a thesis (suggested)
Line 146-152 please combine smaller paragraphs into bigger ones, check this everywhere else in the manuscript
Line 184 “the maximum pile tip resistance is within 20% of the applied loading.” How about past studies what were their findings are they in agreement or contradict your results here? Please discuss
Line 187-188 why it occurs there please explain and support with references
Line 228 “enhance pile-soil interaction” please support this claim with a reference and explain further
Combine lines 226-232 in one big paragraph writing small paragraphs is not good practice in manuscript it makes it look vert fragmented
Line 231-232 “should be paid enough” this is a very vague sentence please discuss and support with references
Line 281 can you compare this with past studies in the open literature did they find similar causes to what you have seen in this work?
In all of the discussion the authors critically analyse their work and use past studies to explain what they found using only one reference! This is not acceptable and authors should be fixed
The results are merely described and is limited to comparing the experimental observation. The authors are encouraged to include a discussion section and critically discuss the observations from this investigation with existing literature.
Author Response
Paper Title:
Experimental study on Deformation and Load Transfer Mechanism of Symmetrical Batter Piles under Vertical Loading
Authors: Kaiyuan Liu 1, Chao Han 2*, Chengshun Xu 3 and Zhibao Nie 4
The authors sincerely thank the reviewers for their encouragement and constructive comments. All comments have been carefully considered and revised in the paper or explained as follows. Moreover, we added a single batter pile (10°) test, in order to further illustrate the influence of the mirror-like pile and cap constraint on the redistribution and weaken of the bending moment.
Comments by the reviewers
Comment 1: Please consider reviewing the abstract and highlight the novelty, major findings and conclusions. What is the research gap did you find from the previous researchers in your field? Mention it properly. It will improve the strength of the article. Line 15 instead of current it is better to say previous.
Response: Thanks for the comment. The abstract has been checked and partially rewritten.
Comment 2: Why did the authors use the 10- and 20-degrees inclination in Table 1? Are they common standard angles or based on past literature or industrial applications or something else?
Response: Thanks for the comment. The pile inclination 10° and 20° represent the typical and relative large inclinations in engineering practice. For most of batter pile foundations, the pile inclination is within 8°~15°. For example, the batter pile foundation of approach bridge (6KM) used in Hangzhou Bay is consists of a vertical pile and nine batter piles, all made of steel pipe [1]. The minimum and maximum pile inclinations are 8° and 12°, with pile length 65~78m. The current largest pile inclination is 30°, which is used in I-10 highway bridge at Pontcharain lake, Los Angeles [2][3],but it is an extreme case. In addition, if the increment of a variable is too small, it is difficult to control in tests, especially in the reduced-scale laboratory tests. Therefore, in this paper, we pick up 10° and 20° as the pile inclinations. The summary of batter pile application in worldwide is shown in Table 1.
[1]Wang R.G. Technological innovation and application of Hangzhou Bay Bridge [M]. Zhejiang science and Technology Press, 2008
[2] Abu-Farsakh M , Souri A , Voyiadjis G , et al. Comparison of Static Lateral Behavior of Three Pile Group Configurations Using Three-Dimensional Finite Element Modeling[J]. Canadian Geotechnical Journal, 2017: cgj-2017-0077.
[3] Souri, Ahmad, Abu-Farsakh, et al. Study of static lateral behavior of battered pile group foundation at I-10 Twin Span Bridge using three-dimensional finite element modeling.[J]. Canadian Geotechnical Journal, 2016.
Table 1 The summary of batter pile application in worldwide
|
No. |
Diameter |
Inclination |
Type |
length |
Application |
|
1 |
1.2 m |
9.5° |
PHC pipe pile |
80m |
The approach of Donghai Bridge China |
|
2 |
0.25 m |
12 |
Steel pipe pile |
50 m |
Connection bridge of Qiongzhou Bay China |
|
3 |
1.5 m |
9.5° |
Steel pipe pile |
88 m |
the approach of Hangzhou Bay Bridge China |
|
4 |
1.8 m |
6 |
Bored pile |
30 m |
River WMS Bridge Germany |
|
5 |
1.2 m |
6 |
Bored pile |
44 m |
Swietokrayshi Bridge poland |
|
6 |
1.2 m |
9 |
Bored pile |
25 m |
Ice high speed railway Munich-Nuremberg Germany |
|
7 |
1.0 m |
9 |
Bored pile |
37 m |
WERN Bridge Germany |
|
8 |
0.9 m |
12 |
Bored pile |
20 m |
Berlin maglev railway Germany |
|
9 |
1.2 m |
9.5~14 |
Concrete pile |
4.5 m |
500kV Gangcheng-Maoming transmission line China |
|
10 |
1.0 m |
9.5~14 |
Bored pile |
45 m |
Petrochemical terminal of Daya Bay China |
|
11 |
1.2 m |
14 |
Steel pipe pile |
80 m |
Fujian Kemen power plant wharf China |
Comment 3: The table have many symbols and letters please create a list of symbols and nomenclature and add it at the start or end of the manuscript.
Response: Thanks for the reminding. The meaning of symbols has been listed as an appendix A in the end of the paper.
Comment 4: Figure 2 is not necessary and can be removed, it is more suitable for a book chapter or literature review chapter in a thesis (suggested)
Response: Thanks for the suggestion. The Figure 2 has been removed.
Comment 5: Line 146-152 please combine smaller paragraphs into bigger ones, check this everywhere else in the manuscript.
Response: Thanks for the suggestion. All manuscript has been checked. If several paragraphs have the same discussing contents, they are merged into a bigger one.
Comment 6: Line 184 “the maximum pile tip resistance is within 20% of the applied loading.” How about past studies what were their findings are they in agreement or contradict your results here? Please discuss.
Response: “The maximum pile tip resistance is within 20% of the applied loading” here just used to classify the pile into floating or end-bearing. The aim is to state that the deformation mode acquired in this paper is based on floating batter pile test. The 20% is not used to make a universal point of view. Actually, the tip resistance ratio varies from tests to tests, projects to projects, it is related to many factors. It cannot be specified unless the test method, soil properties, pile properties, the stiffness ratio pile to soil and other factors are exactly the same.
Comment 7: Line 187-188 why it occurs there please explain and support with references.
Response: Thanks for the comment. When the pile subjected to the horizontal loading, the load will be borne by the lateral impedance of the overall pile-soil system. With the increase of the depth, the lateral load decreases and so the pile-soil interaction. The upper pile section (usually within 0.1-0.5L) has the strongest pile-soil interaction, which is widely agreed with the community[1-4]. Therefore, it is not been explained.
The effective length depends on the ratio of lateral pile to soil stiffness, the boundary condition of pile top et al. Though the batter pile in this paper subjected to the vertical loading, the load will be decomposed to the lateral and axial direction along the pile as illustrated in the new abstract. Therefore, the vertical loaded batter pile will appears lateral pile-soil interaction which is similar to the lateral loaded plumb pile. It is also occurs in the upper part of pile (0.2L in this paper).
[1] Rollins K M , Peterson K T , Weaver T J . Lateral Load Behavior of Full-Scale Pile Group in Clay[J]. Journal of Geotechnical & Geoenvironmental Engineering, 1998, 124(6):468-478.
[2] Rowe, R. K . Pile Foundation Analysis and Design[J]. Canadian Geotechnical Journal, 1980, 18(3):472-473.
[3] Ruprai M S . Behaviour of model flexible piles under inclined loads in sand[J]. Memorial University of Newfoundland, 1987.
[4] Lin H , Ni L , Suleiman M T , et al. Interaction between Laterally Loaded Pile and Surrounding Soil[J]. Journal of Geotechnical & Geoenvironmental Engineering, 2014, 141(4):04014119.
Comment 8: Line 228 “enhance pile-soil interaction” please support this claim with a reference and explain further.
Response: Thanks for the comment. Line 228 just mentioned this phenomenon in order to lead to the discussion in the following. Specific analysis is in the next part as stated in the paper. This conclusion is a reasonable inference from the results of the experiment. The pile deformation are derived from strain of pile, which is the true value. As Figure 1 shown, symmetrical batter pile deforms in the opposite directions, plumb deforms in the same directions. This clamping soil between batter piles subjects to bidirectional compression, the confining pressure of soil will higher than that of the plumb pile. It is well known that the higher the confining pressure, the higher the soil stiffness. With the increases of deformation, the normal interaction force, the friction coefficient will also be increased. The analysis above is the meaning of “enhance pile-soil interaction”. The new analysis is added in the paper.
Figure1
Comment 9: Combine lines 226-232 in one big paragraph writing small paragraphs is not good practice in manuscript it makes it look very fragmented.
Response: Thanks for the suggestion. They has been combined.
Comment 10: Line 231-232 “should be paid enough” this is a very vague sentence please discuss.
Response: Thanks for the comment. “Should be paid enough” has been explained in Comment 8 and its Response.
Comment 11: Line 281 can you compare this with past studies in the open literature did they find similar causes to what you have seen in this work?
Response: Thanks for the comment. Li [1-2] has the similar observation with our tests. But his pile group is consist of vertical and batter piles and made of alumni, which is different with us, that is why we did not mention it in the article. Moreover, his experiments mainly focus on the different functions between plumb and batter pile in one pile group,the “clamping” effect is not analyzed properly. His works has been mentioned in the paper.
[1] Li Longqi,Deng Xiaoxue,Zhang Shuai,et al.Study on vertical bearing capacity and pile group effect of inclinedpile foundation in inhomogeneous strata[J].Advanced Engineering Sciences,2020,52(4):1–11
[2]Li LQ, Luo SX. A simulation test study of vertical bearing capacity of inclined pile foundation in inhomogeneous strata [J]. Rock and soil Mechanics, 2012, 33(005):001300-1305.
Comment 12: In all of the discussion the authors critically analyses their work and use past studies to explain what they found using only one reference! This is not acceptable and authors should be fixed.
Response: Thanks for the comment. As stated in the introduction, at present, few literatures studied the batter pile. Nearly none literature have focus on the different of load transfer mechanism and working behavior between the single and symmetrical batter piles. For a comparison, the test condition should be similar or at least the important factors are the same. Unfortunately, in the published literatures, either the configuration of pile is different or the pile material(pile properties/ the stiffness ratio pile to soil).
In order to solve the problem, we added a new test of single batter pile (β=10°). The new test's bending moment distribution combine with Cao’s test result is capable to illustrate the redistribution and weaken of the bending moment by the mirror-like pile and cap constraint. The reanalysis including the new test has been added in the paper.
Thanks again for the great comments and suggestion from reviewer. Best regards!
Liu KY

Reviewer 2 Report
Dear Sirs
In my opinion your work is formally correct, follows the standard structure of research paper and seem to be well written. My only concern is based on my engineering experience and conviction that symmetrical batter piles are used mainly to provide a load transfer in the case of inclined loads. The pair of piles and their inclination works well when bending moments and lateral forces are applied to the capping beam.
The current study examines vertical load only. Formally it is correct, but I'd appreciate its value only as a preliminary study for further research on inclined loads. My detailed comments are given below:
- Please correct the name of G.G. Meyerhof (letter "f" missing in the reference list [9-12]).
- Referring to multiple references try to provide at least one sentence per reference, explaining its impact on your current work.
- I'd appreciate some discussion (maybe I did not notice it) on "free length" of the batter piles over the ground level. That value may be of similar importance to the pile spacing at the capping beam level.
- Please specify clearly in the conclusions, what are the prospects for your future studies. Presented work, if not continued with regard to inclined load seem to be "stopped half-way".
Sincerely yours
Alexander Ivannikov
Author Response
Paper Title:
Experimental study on Deformation and Load Transfer Mechanism of Symmetrical Batter Piles under Vertical Loading
Authors: Kaiyuan Liu 1, Chao Han 2*, Chengshun Xu 3 and Zhibao Nie 4
The authors sincerely thank the reviewers for their encouragement and constructive comments. All comments have been carefully considered and revised in the paper or explained as follows.
Comments by the reviewers
Comment 1: Please correct the name of G.G. Meyerhof (letter "f" missing in the reference list [9-12]).
Response: Thanks for the reminding, the writing mistake has been corrected.
Comment 2: Referring to multiple references try to provide at least one sentence per reference, explaining its impact on your current work.
Response: Thanks for the suggestion. Multiple referring has been split into individual with comments added to each (in the part of introduction)
Comment 3: I'd appreciate some discussion (maybe I did not notice it) on "free length" of the batter piles over the ground level. That value may be of similar importance to the pile spacing at the capping beam level.
Response:
Thanks for the comment, it is very meaningful and instructive. In terms of practice experience, the increase of the “free length” will increase the pile spacing at the mud-line, resulting in the decrease of the vertical group effect of batter pile. However, the "free length" is often related to the depth of water, the force of wave and current, the values may vary greatly in different engineering practices. Therefore, it is difficult to determine a practical free length in the scale-reduced experiment. In addition, in the action of vertical load, the cap will undergo a downward displacement, which increases the difficulty in discussing the influence of free length on working behavior. Therefore, in this paper, the variation of free length is not take into account.
The tests shown in this paper is the preliminary study on batter pile foundation of offshore wind turbines. A series of static and dynamic experiments will be carried out in the year 2021-2022. As stated in the above, the variable of free length is difficult to control in the experiment. Therefore, in the follow-up, we are preparing to discuss the influence of free length on batter pile's working behavior with numerical method. The tests will be set as the reference and used to validate the numerical model.
Comments 4: Please specify clearly in the conclusions, what are the prospects for your future studies. Presented work, if not continued with regard to inclined load seem to be "stopped half-way".
Response: Thanks for the suggestion. The prospects of future studies are clarified briefly in the end of the paper as:
Actually, these tests have three purposes
- For most of codes in the world, the critical pile spacing without group effect is within 6D-8D for plumb piles. Because of the pile inclination, the batter pile spacing will increases with the increase of depth, namely, the group effect is smaller than plumb pile groups. Based on the principle of stress superposition, the author intends to derive the group effect formula of batter piles by modifying the empirical formula of group effect of plumb piles. Obviously, more tests of plumb piles under the same conditions are needed.
- There are few experiments about the symmetrical batter under vertical loading. In most of the published literatures, important values such as properties of soil and piles are incomplete or ambiguous. The tests presented here will be used to validate the FE model. In this way, the scope of influencing factors can be expanded, the increment of variables can be set more finely. The essential goal is to get the reliable relationship between bearing capacity, pile spacing and pile inclination. Finally to derive the semi empirical formula based on elastic-plastic through fitting
- These tests are the preliminary study for the batter foundation of offshore wind turbines, one of the purpose is to determine the proper range of self-weight of the superstructure. In the following tests, the holistic model with batter pile foundation (four symmetrical batter piles) and high-raised superstructures will be built. The cyclic load will be applied on the top of the superstructure to simulate the wind load. The bearing capacity, the uplift and compress working behavior, the stiffness degradation of the foundation et al. will be explored.
Thanks again for the great comments and suggestion from reviewer. Best regards!
Liu KY

Round 2
Reviewer 1 Report
all questions answered